# Effect of malaria parasite shape on its alignment at erythrocyte membrane

**Anil K Dasanna, Sebastian Hillringhaus, Gerhard Gompper, Dmitry A Fedosov***

Theoretical Physics of Living Matter, Institute of Biological Information Processing and Institute for Advanced Simulation, Forschungszentrum Jülich, Jülich, Germany

**Abstract** During the blood stage of malaria pathogenesis, parasites invade healthy red blood cells (RBC) to multiply inside the host and evade the immune response. When attached to RBC, the parasite first has to align its apex with the membrane for a successful invasion. Since the parasite's apex sits at the pointed end of an oval (egg-like) shape with a large local curvature, apical alignment is in general an energetically unfavorable process. Previously, using coarse-grained mesoscopic simulations, we have shown that optimal alignment time is achieved due to RBC membrane deformation and the stochastic nature of bond-based interactions between the parasite and RBC membrane (Hillringhaus et al., 2020). Here, we demonstrate that the parasite's shape has a prominent effect on the alignment process. The alignment times of spherical parasites for intermediate and large bond off-rates (or weak membrane-parasite interactions) are found to be close to those of an egg-like shape. However, for small bond off-rates (or strong adhesion and large membrane deformations), the alignment time for a spherical shape increases drastically. Parasite shapes with large aspect ratios such as oblate and long prolate ellipsoids are found to exhibit very long alignment times in comparison to the egg-like shape. At a stiffened RBC, a spherical parasite aligns faster than any other investigated shape. This study shows that the original egg-like shape performs not worse for parasite alignment than other considered shapes but is more robust with respect to different adhesion interactions and RBC membrane rigidities.

*For correspondence:
d.fedosov@fz-juelich.de

**Competing interests:** The authors declare that no competing interests exist.

## Introduction

Malaria is a mosquito-borne infectious disease caused by a protozoan parasite of the genus *Plasmodium*. Prior to transmission, the parasite proceeds through both asymptomatic and symptomatic developmental stages in the host (*Miller et al., 2002*; *Cowman et al., 2012*; *White et al., 2014*). After an asymptomatic development stage within the liver, merozoites are released into the bloodstream. They have an egg-like shape with a typical size of approximately 1.5μm (*Bannister et al., 1986b*; *Dasgupta et al., 2017*; *Dasgupta et al., 2014b*). During the blood stage of infection, which is a clinically symptomatic stage, parasites invade healthy red blood cells (RBCs) and multiply inside them. This process aids parasites to evade the immune response. The total life-cycle within each infected RBC lasts for about 48 hr, after which the cell membrane is ruptured and new merozoites are released into the bloodstream.

Invasion of RBCs by parasites is a complex process that involves the following steps: (i) initial random attachment, (ii) reorientation (or alignment) of the apex toward cell membrane, and (iii) formation of a tight junction followed by the final invasion (*Koch and Baum, 2016*; *Cowman and Crabb, 2006*). The parasite's apex contains the required machinery for the invasion process, and thus, apex alignment toward the cell membrane is a necessary step for a successful invasion to follow. Merozoite adhesion to a RBC is facilitated by proteins at the parasite surface which can bind to the cell membrane (*Bannister et al., 1986b*; *Gilson et al., 2006*; *Beeson et al., 2016*). Recent optical tweezers experiments provide an estimation for the force required to detach a parasite adhered to RBC membrane to be in the range of 10–40 pN (*Crick et al., 2014*). Other experiments (*Dvorak et al.,*

*1975*; *Gilson and Crabb, 2009*; *Glushakova et al., 2005*; *Crick et al., 2013*) demonstrate that the parasite is dynamic at the RBC membrane, and induces considerable membrane deformation during alignment. Furthermore, there is a positive correlation between such deformations and parasite alignment. The time required for the parasite to align is found to be on the order of 16 s (*Weiss et al., 2015*). Our recent investigation of the parasite alignment with adhesion modeled by a homogeneous interaction potential has confirmed the importance of membrane deformations for proper alignment, but the alignment times were found to be significantly less than 1 s (*Hillringhaus et al., 2019*). The main shortcoming of this model is that it produces only static membrane deformations and the parasite exhibits very little dynamics at the RBC surface. This model has been extended by including realistic bond-based adhesion interactions between the parasite and RBC membrane (*Hillringhaus et al., 2020*), which results in alignment times consistent with the experimental measurements.

A typical merozoite has an egg-like shape with the apical complex sitting on the pointed edge. Our previous work (*Hillringhaus et al., 2020*) suggests that parasite alignment occurs due to RBC deformability and stochastic fluctuations in bond dynamics. Stochastic fluctuations and consequent rolling-like (or rotational) motion of the parasite at the membrane surface are especially important at low adhesion strengths, as they facilitate alignment toward pointed apex. The egg-like shape naturally adheres to RBC membrane with its less curved side, as this adhesion state corresponds to the largest contact area. Then, a rotational motion of the parasite toward the apex is required to establish an apex-membrane contact. If parasite adhesion interactions with a membrane are strong, merozoite mobility is significantly suppressed, and the alignment is mainly facilitated through wrapping of the parasite by cell membrane, emphasizing the importance of RBC deformability. These are two major mechanisms for the alignment of an egg-like merozoite.

Even though most types of *Plasmodium* merozoites have an egg-like shape, the merozoite of *Plasmodium yoelii* changes its shape from an oval to a spherical shape right before its attachment followed by alignment and invasion (*Yahata et al., 2012*). The alignment time for *Plasmodium yoelii* is also reported to be longer than for *Plasmodium falciparum* (*Yahata et al., 2012*). It is not clear why the *Plasmodium yoelii* parasite adapts its shape before the alignment process at the RBC membrane. This raises a question whether the parasite shape has important advantages/disadvantages in the alignment process or simply results from the structural organization of its internal elements. Therefore, it is important to understand the effect of parasite shape in the alignment process.

In this article, the role of parasite shape in the alignment process at the RBC membrane is studied by mesoscopic computer simulations. In particular, we show that basic dynamical properties, such as parasite mobility, RBC membrane deformation, and the number of adhesion bonds, are significantly affected by different parasite shapes. In turn, these are tightly coupled to parasite alignment characteristics that determine its alignment success. In general, parasite shapes with large aspect ratios (e.g. oblate and long prolate ellipsoid) are disadvantageous for alignment, as these shapes result in a significant reduction of parasite mobility at the membrane. A spherical parasite is more mobile than an egg-like merozoite, which is advantageous in cases with low adhesion interactions or increased membrane stiffness. However, the spherical shape is disadvantageous for strong adhesion interactions, when parasite mobility is suppressed, as parasite alignment by membrane wrapping is often unsuccessful because the apex may not be within the adhesion area. As a result, the egg-like shape exhibits an alignment performance that is generally not worse than of other studied shapes, but more robust for disparate conditions in parasite adhesion strength and RBC membrane deformability.

## Results

To investigate the role of parasite's shape in the alignment process, five different shapes with varying aspect ratios are chosen: (*i*) an egg-like (EG) shape that is the typical shape of *Plasmodium falciparum* merozoite, (*ii*) a sphere (SP), (*iii*) a short ellipsoid (SE) whose dimensions are similar to the egg-like shape, (*iv*) a long ellipsoid (LE), and (*v*) an oblate (OB) shape, see *Figure 1(a)*. The corresponding maximum and minimum dimensions for these shapes are $r_{\max} = 1.5\,\mu m$ & $r_{\min} = 1.08\,\mu m$ for EG, $r_{\max} = r_{\min} = 1.2\,\mu m$ for SP, $r_{\max} = 1.6\,\mu m$ & $r_{\min} = 1.02\,\mu m$ for SE, $r_{\max} = 2.4\,\mu m$ & $r_{\min} = 0.76\,\mu m$

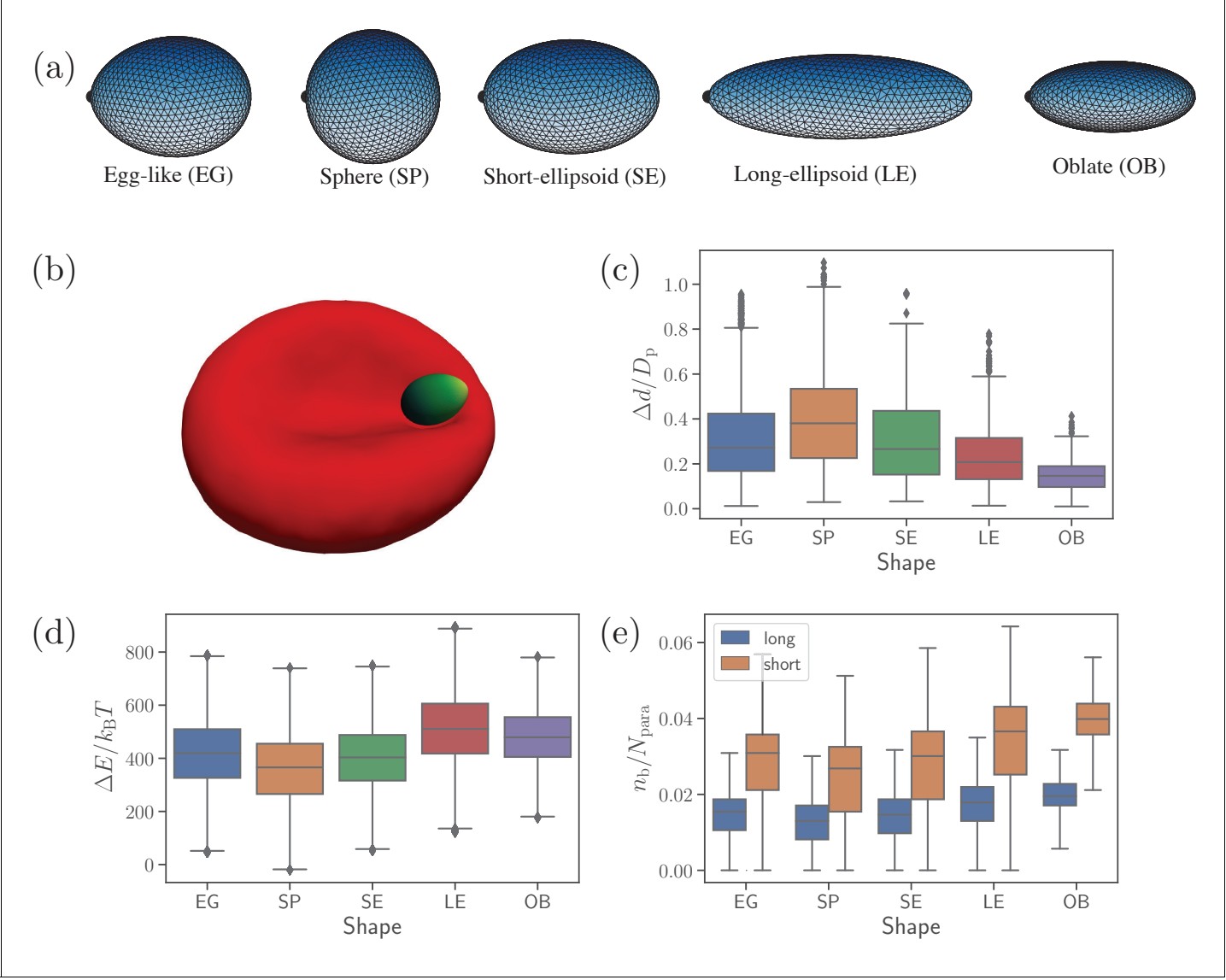

**Figure 1.** Different parasite shapes and their dynamic properties. (**a**) Triangulated surfaces of different parasite shapes including an egg-like (EG) shape, a sphere (SP), a short ellipsoid (SE), a long ellipsoid (LE), and an oblate ellipsoid (OB). The apex position is indicated by a black point for all parasite shapes. (**b**) A snapshot from simulations showing an egg-like parasite interacting with the RBC membrane, see also **Video 1**. A bright yellow color indicates the apical complex and a dark green color represents the parasite's back. The egg-like (EG) shape is a typical shape of merozoites. (**c**–**e**) Different dynamical characteristics for various parasite shapes. (**c**) Fixed-time displacement $\Delta d$ of the parasite normalized by an effective parasite diameter $D_p = \sqrt{A_p/\pi}$ where $A_p$ is the parasite membrane area. (**d**) Change in total membrane energy $\Delta E$ due to deformation induced by the parasite. (**e**) Number of short and long bonds $n_b$. In (**c**)-(**d**), all data are for the reference parameter set, see **Table 1**.
The online version of this article includes the following source data for figure 1:

**Source data 1.** Source data for graphs shown in **Figure 1(c–e)**.

for LE, and $r_{\max} = 1.5\,\mu m$ & $r_{\min} = 0.64\,\mu m$ for OB. All shapes are selected such that they have approximately the same surface area and the same number of vertices or equivalently the same density of adhesion receptors. The fraction of receptors that can form long bonds is kept at 0.4 similarly to our previous study (**Hillringhaus et al., 2020**), while the fraction of receptors for short bonds is equal to 0.6. Other parasite-RBC interaction parameters are calibrated through the displacement of an egg-shaped parasite at RBC membrane (see **Figure 1(b)** and **Video 1**) against available experimental data (**Weiss et al., 2015**). In this calibration procedure, kinetic rates and strength of the bonds are adjusted, so that simulated and experimental translational displacements of the parasite

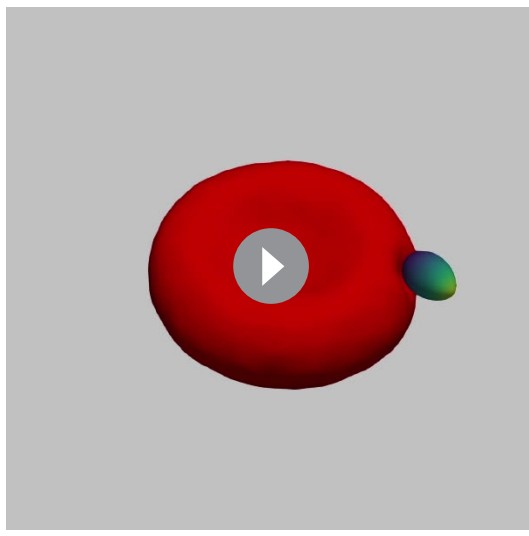

**Video 1.** Motion of an egg-shaped parasite at the membrane of a deformable RBC for the reference RBC-parasite interactions. $k_{\text{off}}/k_{\text{on}}^{\text{long}} = 2$.
https://elifesciences.org/articles/68818#video1

match well, see *Table 1* and the Materials and methods section for details. Note that the translational displacement of the parasite results from its stochastic rolling-like motion. The set of calibrated parameters in *Table 1* will be referred to as the reference parameter set. All other parameters including bond rates are kept same unless stated otherwise.

## Dynamical properties of different shapes

*Figure 1(c-e)* presents basic dynamical measures of merozoites with different shapes. These include [*Figure 1(c)*] fixed-time displacements $\Delta d$ traveled by the parasite over fixed time intervals of $\Delta t = 1\,\text{s}$ and normalized by an effective parasite diameter $D_p = \sqrt{A_p/\pi}$ ($A_p \simeq 4.6\,\mu m^2$ is the parasite membrane area for all shapes), [*Figure 1(d)*] change in membrane total energy $\Delta E/k_{\text{B}}T$ due to deformation induced by parasite adhesion, and [*Figure 1(e)*] the number of bonds $n_{\text{b}}/N_{\text{para}}$. The spherical shape is most mobile (i.e. has the largest $\Delta d/D_p$, see *Video 2*), while the oblate shape is slowest with the lowest $\Delta d$. Intuitively, a shape with a lower local curvature (e.g. OB shape) should form a larger adhesion area, and thus be less dynamic or mobile. This is in agreement with our results in *Figure 1(e)*, where the SP (OB) shape has the smallest (largest) number of bonds, which is directly proportional to the adhesion area. Note that the egg-like and short-ellipsoid shapes show very similar dynamic characteristics, as these shapes are very close to each other. Fixed-time displacement of the long ellipsoid (see *Videos 2* and *3*) has values between those for the SE and OB shapes, which is consistent with the number of bonds (or equivalently the adhesion area) in *Figure 1(e)*.

The RBC deformation energy $E/k_{\text{B}}T$ can generally be expected to be proportional to the adhesion area or the number of formed bonds. This is true for the spherical shape that induces the lowest deformation energy in *Figure 1(d)*. Furthermore, both oblate and long ellipsoid shapes result in a large deformation energy. However, the LE shape has a slightly larger value of $\Delta E$ than the OB shape, even though the oblate shape has a larger adhesion area. This can be rationalized by the fact that the adhesion of the oblate shape to RBC membrane induces a lower deformation than the LE

**Table 1.** List of kinetic bond parameters that are used to calibrate the parasite's translational displacement in simulations against experimental data by *Weiss et al., 2015*.
$\tau = \eta D_0^3/\kappa$ is the membrane relaxation timescale.

| Parameter | Simulation value | Physical value |
|---|---|---|
| $\ell_{\text{ext}}^{\text{long}}$ | $0.0154\,\text{D}_0$ | 100 nm |
| $\ell_{\text{ext}}^{\text{short}}$ | $0.0031\,\text{D}_0$ | 20 nm |
| $\rho_{\text{long}}$ | $0.4\,\rho_{\text{para}}$ | 107 µm$^{-2}$ |
| $\rho_{\text{short}}$ | $0.6\,\rho_{\text{para}}$ | 161 µm$^{-2}$ |
| $k_{\text{on}}^{\text{long}}$ | $36.3\,\tau^{-1}$ | 39.6 s$^{-1}$ |
| $k_{\text{on}}^{\text{short}}$ | $290.3\,\tau^{-1}$ | 317.0 s$^{-1}$ |
| $k_{\text{off}}$ | $72.58\,\tau^{-1}$ | 79.2 s$^{-1}$ |
| $\lambda_{\text{long}}$ | $2.46 \times 10^4\,\text{k}_{\text{B}}\text{T}/\text{D}_0^2$ | 2.57 µN m$^{-1}$ |
| $\lambda_{\text{short}}$ | $0.82 \times 10^4\,\text{k}_{\text{B}}\text{T}/\text{D}_0^2$ | 0.856 µN m$^{-1}$ |

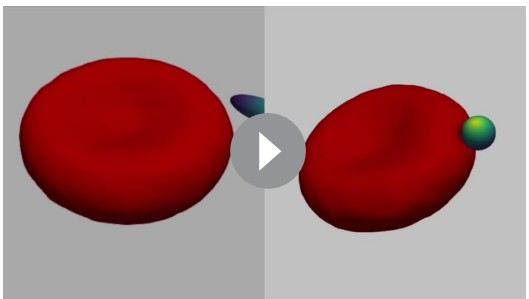

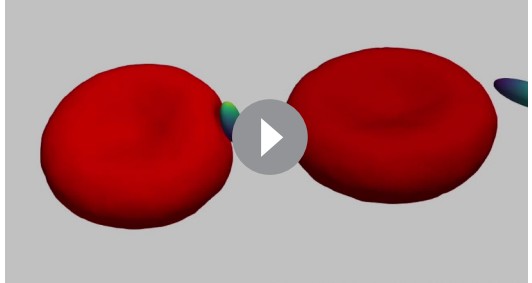

**Video 2.** Mobility of the LE- and SP-shaped parasites at the RBC membrane, which can be compared to the EG shape in **Video 1**. Here, $k_{\text{off}}/k_{\text{on}}^{\text{long}} = 2$.
https://elifesciences.org/articles/68818#video2

**Video 3.** Mobility of the LE- and OB-shaped parasites at the RBC membrane, which can also be compared to the EG shape in **Video 1**. Here, $k_{\text{off}}/k_{\text{on}}^{\text{long}} = 2$.
https://elifesciences.org/articles/68818#video3

shape, as the OB shape has a lower curvature at its flat side (see *Video 3*). Furthermore, the exact position where parasite adheres to RBC membrane is important, as local curvature at the membrane (e.g. negative curvature in the dimple areas) can oppositely match the curvature at the parasite surface. This is the main reason why a successful alignment occurs more frequently in the concave areas of RBC dimples than at the convex rim of the membrane (*Hillringhaus et al., 2020*). Note that the RBC deformation energy in *Figure 1(d)* displays opposite trends in its shape dependence than the parasite mobility or fixed-time displacement in *Figure 1(c)*. As a result, parasite shapes with a large asphericity are less dynamic at the RBC surface, while the EG, SP, and SE shapes show comparable dynamical characteristics.

## Parasite alignment characteristics

To characterize parasite alignment, we introduce two quantities: (*i*) apex distance $d_{apex}$ and (*ii*) alignment angle θ (*Hillringhaus et al., 2020*) given by

$$d_{\text{apex}} = \min_i(|\mathbf{r}_{\text{apex}} - \mathbf{r}_i|), \qquad \theta = \arccos(\mathbf{n} \cdot \mathbf{n}^{\text{face}}), \tag{1}$$

where $\mathbf{r}_{\text{apex}}$ is the apex position, $\mathbf{r}_i$ is the position of vertex $i$ at the membrane, $\mathbf{n}$ is the parasite's directional vector pointing from its back to the apex, and $\mathbf{n}^{\text{face}}$ is the normal vector of a RBC membrane triangular face whose center of mass is closest to the parasite's apex. Both the apex distance $d_{apex}$ and the alignment angle θ are schematically depicted in *Figure 2(a)*. The directional vector is defined for all shapes by selecting two opposite vertices along the shape axis, which represent the apex and the back. A perfect alignment is achieved when the apex distance is equal to the distance at which the repulsive interaction vanishes (i.e. at $\simeq 2^{1/6}\sigma$) and the alignment angle θ is equal to π. Due to limitations in the discretization of both the RBC and the parasite (*Hillringhaus et al., 2020*), a successful parasite alignment can be characterized by the criteria

$$d_{\text{apex}} \leq 2^{1/6}\sigma + r_{\text{junc}} \quad \& \quad \theta \geq 0.8\pi, \tag{2}$$

where $r_{\text{junc}} = 10$ nm defines the junctional interaction range of the parasite's apex (*Bannister et al., 1986b*).

*Figure 2* presents apex-distance and alignment-angle distributions for different parasite shapes, where the alignment criteria from Eq (2) are indicated by the dashed lines. Even though alignment of the SE shape is similar to the EG shape, it is slightly worse for the SE shape as the $d_{apex}$ distribution in *Figure 2(a)* is shifted further away from the alignment criterion for $d_{apex}$ than that for the EG shape. This is due to the fore-aft asymmetry of the EG shape, whose largest-adhesion configuration corresponds to a slightly tilted orientation of the parasite with its apex closer to the membrane than its back. *Figure 2(b) and (d)* compares the alignment characteristics of the egg-like shape with oblate and long-ellipsoid shapes. The key advantage of the egg-like shape in comparison to LE and OB shapes is that the EG shape has a reduced adhesion area due to a larger local curvature, which allows the EG parasite to fluctuate more around its directional vector, leading to wider distributions

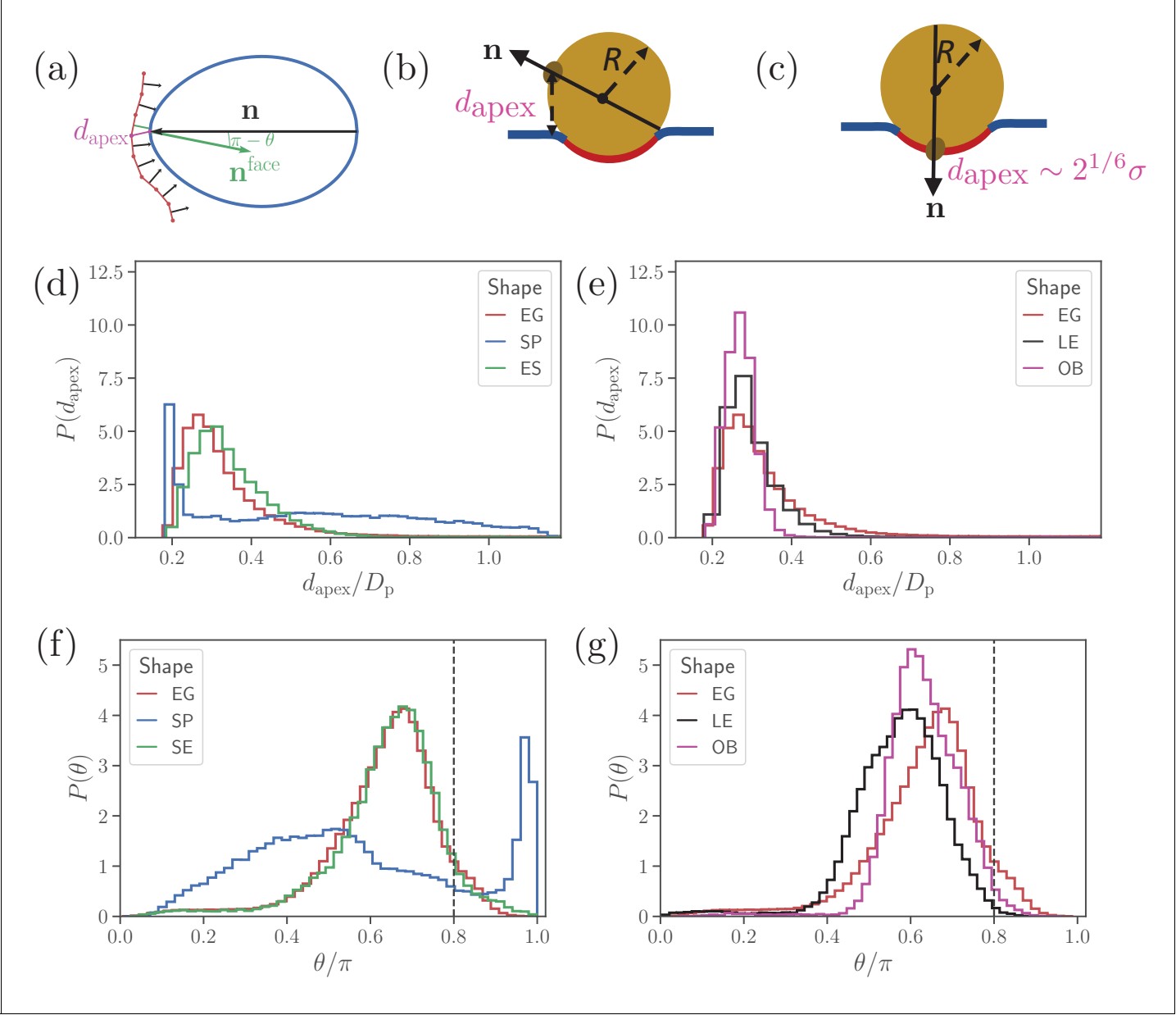

**Figure 2.** Parasite alignment characteristics. (a) Sketch of parasite at the RBC membrane. Alignment angle, θ, is defined as the angle between the parasite's directional vector n (black arrow) and membrane surface normal $n^{face}$ (green arrow). The apex distance, $d_{apex}$, is the distance between the parasite's apex and the RBC membrane surface. (b) and (c) Sketch of a spherical parasite of radius $R$, partially wrapped by a membrane area $A_m$, with its apex (b) away from the adhesion area and (c) within the wrapped area. (d) and (e) Apex distance $d_{apex}$ distributions and (f) and (g) alignment angle θ distributions for all shapes. In all plots, the alignment criteria from *Equation (2)* are shown by the dashed lines. For a better readability, the distributions for SP and SE shapes in (d) and (e) are plotted separately from the distributions for LE and OB shapes in (f) and (g) along with distributions of the egg-like shape (EG) in all plots.

The online version of this article includes the following source data for figure 2:

**Source data 1.** Source data for graphs shown in *Figure 2(d–g)*.

of its alignment characteristics. Since the OB shape forms the largest adhesion area in comparison to the EG and LE shapes, it has the narrowest distributions (i.e. lowest fluctuations) for both the apex distance and alignment angle. A further advantage of the EG shape is the aforementioned fore-aft asymmetry, which results in a shift of the apex-distance and alignment-angle distributions toward a better alignment in comparison to those for the OB and LE shapes.

Clearly, parasite alignment characteristics for the spherical shape are qualitatively different from all other (ellipsoid-like) shapes. At first glance, $d_{apex}$ and θ distributions in *Figure 2(d)* and (f) seem to suggest that the SP shape might be best for parasite alignment. However, the membrane deformability breaks the 'up-down' symmetry of these distributions, leading to two distinct cases: (*i*) the apex is not within the parasite-membrane adhesion area [*Figure 2(b)*] for which the alignment characteristics are very poor and (ii) the apex is within the adhesion area [*Figure 2(c)*] resulting in good alignment. Hence, alignment performance of the spherical parasite is more subtle than indicated by the distributions in *Figure 2* and will be discussed further below.

## Alignment of a spherical parasite

*Figure 3* shows apex-distance and alignment-angle distributions of the spherical parasite for different values of the off-rate $k_{off}$. Both $d_{apex}$ and θ distributions generally display a sharp peak near the alignment criteria and a long tail with a wider distribution for non-aligned parasite orientations. Thus, the sharp peak represents parasite orientations when its apex is within the membrane-parasite contact area $A_\mathrm{m}$, as schematically illustrated in *Figure 2(c)*. Correspondingly, the long tail characterizes orientations when the parasite's apex is not within the adhesion area $A_\mathrm{m}$, as sketched in *Figure 2 (b)*.

At large values of the off-rate (e.g. $k_{\mathrm{off}}/k_{\mathrm{on}}^{\mathrm{long}} = 4.0$), the spherical parasite is very mobile at the membrane surface (i.e. has a large effective rotational diffusion) and induces nearly no membrane deformations. In such cases, there is no significant wrapping of the parasite by the membrane, resulting in wide $d_{apex}$ and θ distributions in *Figure 3*. As $k_{off}$ is decreased, the merozoite becomes partially wrapped by the membrane, leading to the development of the sharp peak in both distributions. At the smallest off-rate of $k_{\mathrm{off}}/k_{\mathrm{on}}^{\mathrm{long}} = 0.5$, the alignment properties in *Figure 3* seem to be qualitatively different from those for larger off-rates. Note that for small off-rates, the parasite forms a large number of bonds with the membrane, resulting essentially in its arrest with nearly zero rotational diffusion. Therefore, these simulations are too short to fully capture $d_{apex}$ and θ distributions, which are also expected to have the sharp peak characterized by the wrapped area $A_\mathrm{m}$.

To rationalize apex-distance and alignment-angle distributions for a spherical parasite, we use a simple model of a sphere with radius $R$ partially wrapped by the membrane, as illustrated in *Figure 2 (b) and (c)*. Since the adhered parasite is mobile, the parasite's directional vector can point toward any possible direction when sampled over times longer than a characteristic time of parasite

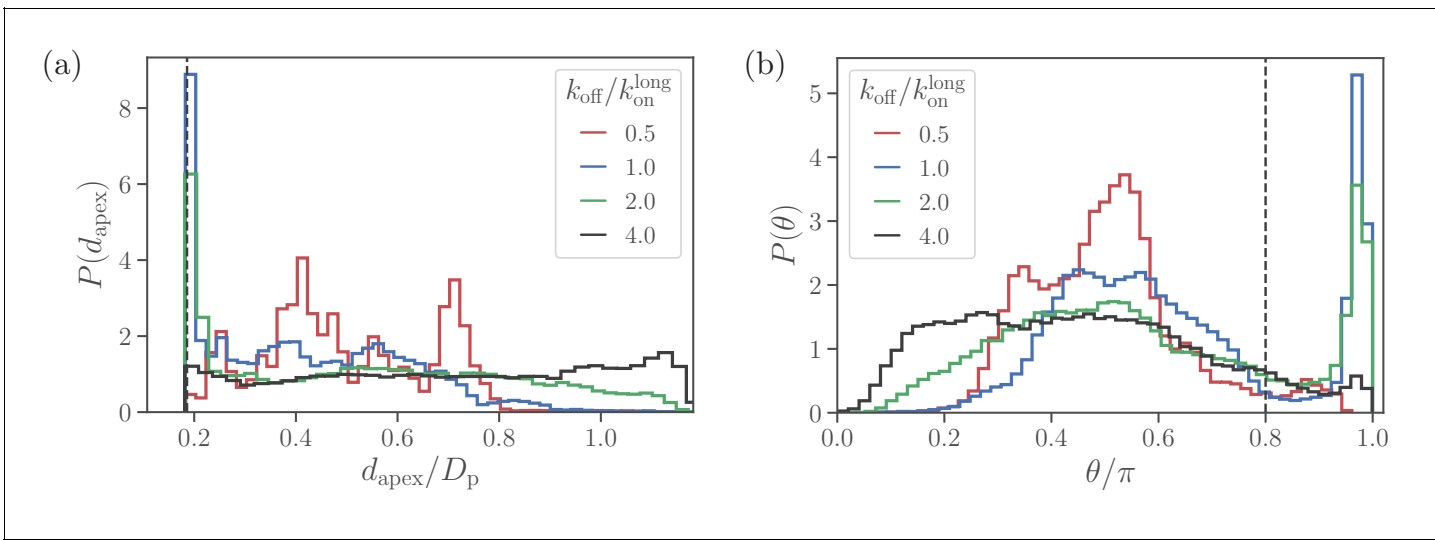

**Figure 3.** Alignment of a spherical parasite. (a) Apex-distance and (b) alignment-angle distributions of the SP shape for different bond off-rates. The alignment criteria from *Equation 2* are shown by the dashed lines. Several peaks in $P(d_{\mathrm{apex}})$ for $k_{\mathrm{off}}/k_{\mathrm{on}}^{\mathrm{long}} = 0.5$ are due to limited statistics, since for the strongest adhesion, the parasite mobility is very low and the simulations are too short to capture the stationary distribution.

The online version of this article includes the following source data for figure 3:

**Source data 1.** Source data for graphs shown in *Figure 3*.

rotational motion. Therefore, the probability of alignment can be approximated as $A_m/A_p$, where $A_m$ and $A_p$ are the adhesion area and the total surface area of a sphere, respectively. As a result, the sharp peak in $d_{apex}$ and $\theta$ distributions near the alignment criteria must increase with an increase in adhesion strength or a decrease in $k_{off}$. A theoretical model for sphere wrapping based on energy minimization results in *Hillringhaus et al., 2019*,

$$\frac{A_m}{A_p} = 2\sqrt{\frac{2}{Y}\left(\frac{\Delta U_b}{A_c} - \frac{2\kappa}{R^2}\right)}, \tag{3}$$

where $Y$ is the Young's modulus of the membrane, $\kappa$ is the bending rigidity, and $\Delta U_b \sim k_B T \ln(k_{on}/k_{off})$ is the energy gained through single-bond association per area $A_c$, which can be considered as the effective area of a single bond. Therefore, $A_m$ increases with a decrease in $k_{off}$. Furthermore, for small deformations, $A_m$ is essentially governed by the competition of bending and adhesion energies, while for strong adhesion, stretching elasticity of the membrane also becomes important.

When the parasite is not aligned, the apex distance can be approximated by a height of the parasite's apex with respect to the flat part of the membrane, see *Figure 2(b)*. Note that this assumption becomes strictly valid for a flat membrane without wrapping or a weak adhesion of the merozoite. Then, the probability distribution for the apex distance $d_{apex}$ is characterized by the area of a spherical segment (or frustum) with height $\Delta h$ as

$$P(d_{apex})\Delta h = \frac{A_{\Delta h}}{A_p} = \frac{2\pi R \Delta h}{A_p} = \frac{1}{2R}\Delta h, \tag{4}$$

where $\Delta h$ is an infinitesimally small interval around $d_{apex}$. Therefore, $P(d_{apex}) = 1/(2R)$ is independent of the apex distance, which is consistent with nearly flat $d_{apex}$ distributions for $k_{off}/k_{on}^{long} \geq 2.0$ in *Figure 3(a)* and *Figure 2(d)*. Similarly, the probability distribution $P(\theta)$ can be approximated using segment areas $A_{\Delta\theta} = 2\pi R^2(\cos(\theta) - \cos(\theta + \Delta\theta)) \approx 2\pi R^2 \sin(\theta)\Delta\theta$, resulting in

$$P(\theta) \approx \frac{\sin(\theta)}{2} \tag{5}$$

for not aligned parasite orientations. This approximation is consistent with the data in *Figure 3(b)* and *Figure 2(f)*. In summary, such unique distributions of alignment properties for the SP shape are possible due to the spherical symmetry. For non-spherical parasite shapes, a sharp peak disappears because parasite adhesion to the membrane favors a specific parasite orientation.

## Effect of adhesion strength on parasite alignment time

*Figure 4(a) and (b)* show apex-distance and alignment-angle properties for different parasite shapes and various off-rates which is used to control the strength of merozoite adhesion to the membrane. The apex distance decreases when the off-rate is decreased or the strength of adhesion is increased. Similarly, the alignment angle increases toward the alignment criterion in *Equation (2)*, as the adhesion strength is increased. For all non-spherical shape cases, successful alignment is generally achieved at low enough $k_{off}$ values, which imply strong membrane deformations and a significant wrapping of the parasite by the membrane. This is consistent with deformation energies shown in *Figure 4(c)*, which significantly increase with decreasing $k_{off}$. The main difference for the spherical parasite is that the best alignment is achieved for intermediate values of off-rates (e.g. $k_{off}/k_{on}^{long} \approx 1.0$). As mentioned before, small values of $k_{off}$ significantly suppress parasite mobility, which is required for successful alignment of the spherical parasite because its apex may not be immediately within the parasite-membrane contact area after initial adhesion. Interestingly, the OB shape results in a significantly lower deformation energy than other merozoite shapes for $k_{off}/k_{on}^{long} < 2.0$. Here, the magnitude of local curvature has a pronounced effect, such that the OB shape forms a large adhesion area over its nearly flat part with very low curvature, while close to the rim, where the curvature is large, adhesion interactions are too weak to induce membrane wrapping and deformation. For the other shapes, the adhesion strength is still sufficient to induce partial wrapping of the parasite by the membrane over moderate curvatures.

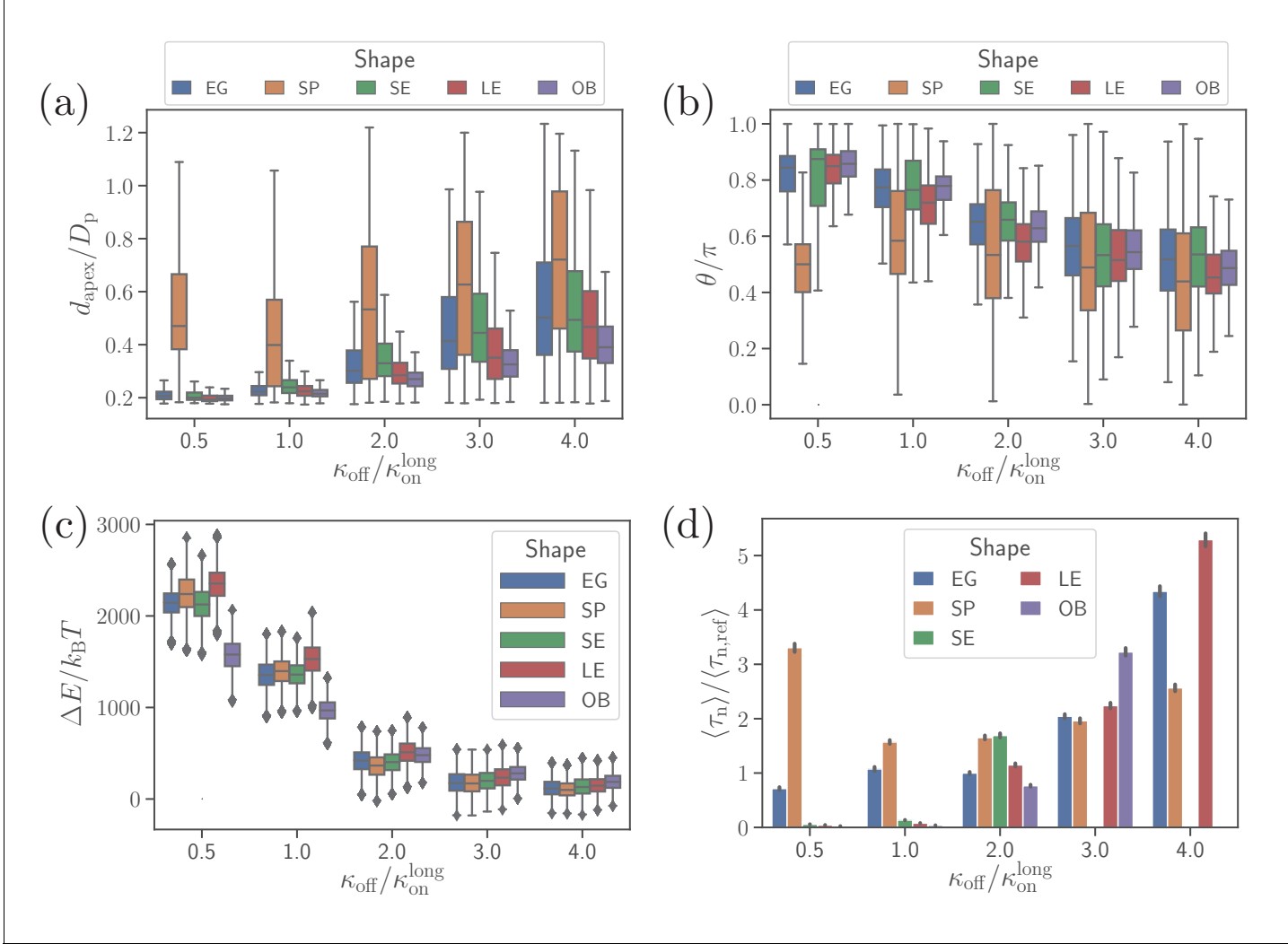

**Figure 4.** Effect of parasite adhesion strength of the alignment time. (a) Apex distance $d_{apex}$, (b) alignment angle $\theta$, (c) total deformation energy $\Delta E$, and (d) alignment time $\tau_{\mathrm{n}}$ for different parasite shapes and bond off-rates that determine the adhesion strength. Several missing bars in the plot of alignment times for $k_{\mathrm{off}}/k_{\mathrm{on}}^{\mathrm{long}}>2$ indicate that $\tau_{\mathrm{n}}$ is much larger than 26 s which is the maximum time of all simulation trajectories.

The online version of this article includes the following source data for figure 4:

**Source data 1.** Source data for graphs shown in *Figure 4*.

To compute alignment times, we employ Monte Carlo simulations based on $(d_{\mathrm{apex}}, \theta)$ probability maps constructed from approximately 10 independent direct simulation trajectories for each parameter set (*Hillringhaus et al., 2020*). *Figure 4(d)* shows alignment times $\tau_{\mathrm{n}}$ for different parasite shapes and off-rates, where all times are normalized by the alignment time of an egg-like shape for the reference parameter set (*Hillringhaus et al., 2020*). In some cases, the bars are missing in the plot, indicating that the alignment has not occurred in direct simulations whose maximum time length is about 26 s. Alignment times of the spherical parasite are very long at small off-rates and become comparable with those of the egg-like shape at intermediate and high values of $k_{off}$. The SE, LE, and OB shapes generally align very fast at small off-rates, but often do not align at all when adhesion becomes weak. This means that these spheroidal shapes require substantial membrane deformation for a successful alignment.

## Alignment at a rigid RBC

To understand the importance of RBC deformability in the alignment process for different parasite shapes, we have simulated parasite alignment at a rigid RBC. *Figure 5(a) and (b)* presents fixed-

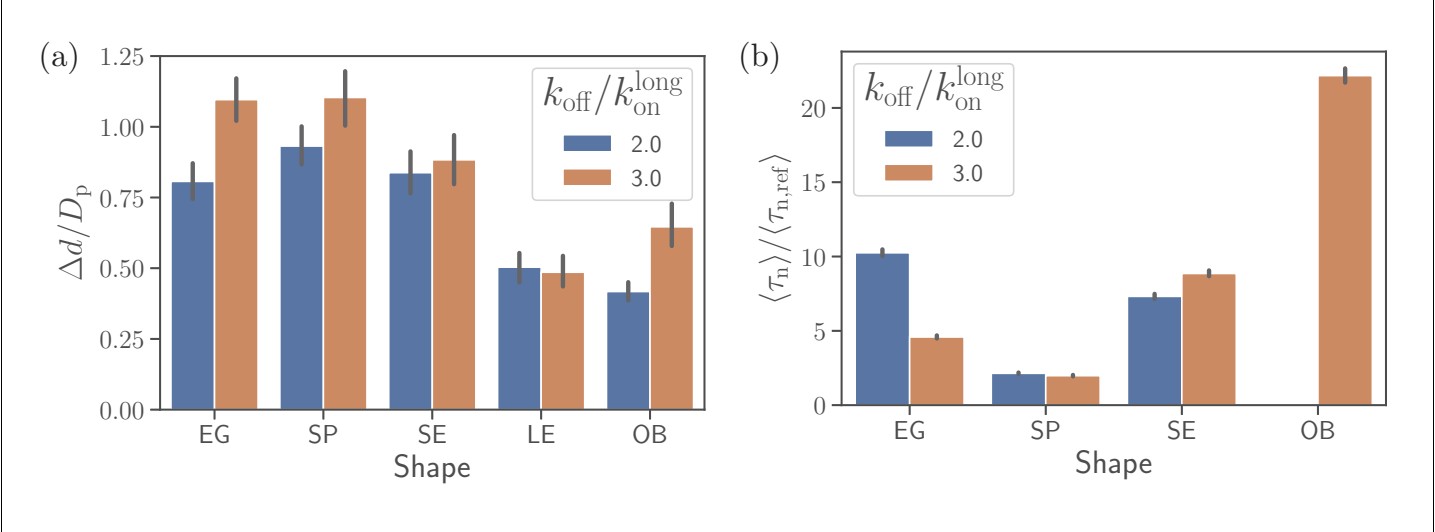

**Figure 5.** Alignment at a rigid membrane. (a) Fixed-time displacement $\Delta d/D_p$ and (b) alignment time $\langle \tau_n \rangle / \langle \tau_{n,\text{ref}} \rangle$ for different parasite shapes and two values of off-rates $k_{\text{off}}/k_{\text{on}}^{\text{long}}$. Note that the data for long ellipsoids is omitted as they never become aligned during direct simulations indicating that their alignment time is much larger than the total simulation time.

The online version of this article includes the following source data for figure 5:

**Source data 1.** Source data for graphs shown in *Figure 5*.

time displacement and alignment time for different parasite shapes and two off-rates. Generally, a small fixed-time displacement (or low mobility at the membrane) results in a long alignment time and vise versa. Both long-ellipsoid and oblate shapes do not align or have a very long alignment time at a rigid membrane, as they require considerable amount of membrane deformation for the alignment. Both egg-like shape and short ellipsoid exhibit similar fixed-time displacements and alignment times. However, for the egg-like shape, an increase in off-rate (i.e. more mobility) improves the alignment whereas for the short ellipsoid an opposite trend is observed. The spherical parasite shows the fastest alignment in comparison to the egg-like and short ellipsoid shapes due to its increased mobility. Thus, at a rigid RBC, the spherical shape shows best alignment properties, at least for intermediate off-rate values.

## Discussion

We have investigated the importance of merozoite shape for its alignment at the RBC membrane which is a prerequisite for the invasion process. This study is a continuation of our previous work (*Hillringhaus et al., 2020*) where the alignment of an egg-like parasite, a natural shape of merozoite, was investigated. Motivated by experimental observations by *Bannister et al., 1986a*, adhesion between the RBC membrane and the parasite is implemented by discrete bonds of two different types, with long and short interaction ranges. The density of both long and short bonds, their kinetic rates and extensional rigidities are calibrated through fixed-time displacement of an egg-like shaped parasite against available experimental data (*Weiss et al., 2015*). Alignment times from two independent experiments are found to be 16 s (*Weiss et al., 2015*) and 7-44 s (*Yahata et al., 2012*), respectively. For the egg-like shape, an average alignment time of $10\,s$ was obtained in our simulations (*Hillringhaus et al., 2020*).

To study the effect of parasite shape on alignment, five different parasite shapes, including the original egg-like (EG) shape, short ellipsoid (SE), sphere (SP), long ellipsoid (LE), and oblate (OB) shapes, were considered. The question 'Which parasite shape performs best for apex alignment and potential invasion?' does not have a unique answer, as parasite performance also depends on the membrane properties and the characteristics of the adhesion bonds, including their dynamics. In general, some parasite shapes are advantageous when the binding kinetics are slow or large RBC

membrane deformations take place, while other shapes are advantageous in case of fast binding kinetics or small membrane deformations.

One of our key results is that the spherical parasite exhibits apex-distance and alignment-angle distributions different from those for non-spherical shapes. Distributions of alignment characteristics for the SP shape generally have a sharp peak near the alignment criteria representing parasite orientations with its apex within the adhesion area. At small off-rates or for strong adhesion interactions, there is a considerable parasite wrapping by the membrane and parasite mobility is suppressed. In this case, a successful alignment of the spherical parasite occurs only if the apex ends up directly within the wrapped part of the membrane. This means that the SP shape exhibits 'all or nothing' alignment behavior at small off-rates. In contrast, the egg-like shape adheres with its side to the membrane, and is able to establish a direct membrane-apex contact due to significant parasite wrapping. On the other hand, the SP shape performs better than the EG shape at large values of off-rates when membrane deformation is almost negligible. The spherical symmetry of the SP shape results in its faster mobility in comparison with the egg-like shape. Furthermore, a fluctuation of the EG parasite toward successful alignment due to adhesive dynamics is associated with a larger energetic barrier in comparison to the SP shape for which all directions of motion are statistically equivalent. This is also the main reason why the spherical shape leads to the fastest alignment at a rigid membrane in comparison to all other shapes. Interestingly, even though most types of *Plasmodium* merozoites have an egg-like shape, *Plasmodium yoelii* transforms into a spherical shape from an egg-like shape after the egress from an infected RBC (*Yahata et al., 2012*). This shape transition seems to be essential for the successful invasion. A plausible hypothesis is that *Plasmodium yoelii* exhibits a rather weak adhesion to RBCs in comparison to *Plasmodium falciparum*, so that the spherical shape becomes advantageous for the alignment process. However, we cannot exclude the possibility that this shape transition is just a result of some internal processes such as cytoskeleton rearrangement, in preparation of the merozoite for a subsequent invasion.

Short ellipsoid geometrically resembles the EG shape except that the egg-like shape has asymmetric ends along the cylindrical axis. Therefore, the SE shape shows alignment characteristics that are closest to the EG shape. However, the alignment of the SE shape is slightly worse than that of the egg-like parasite, as the alignment-angle distribution for short ellipsoid is shifted further away from the alignment criterion in comparison to the egg-like shape. This is due to the asymmetry of EG shape along the cylindrical axis, which favors an adhesion orientation tilted toward the apex. At low enough bond off-rates or strong adhesion interactions, alignment of the SE shape is faster than the egg-like shape, as the wrapping of more curved apex region of the EG parasite is slightly less energetically favorable than that of a symmetric SE shape. Furthermore, alignment of oblate and long-ellipsoid shapes proceeds only through significant wrapping of the parasite by the membrane, which occurs only at low off-rates or for strong adhesion interactions. This is also evident from simulations at a rigid RBC, where alignment times of the LE and OB shapes are either very long or no alignment potentially occurs. Despite the fact that the egg-like shape has some advantages over the investigated spheroidal shapes, it is not clear whether this asymmetry exists simply due to the internal parasite structure (e.g. placement of essential organelles of different sizes) or has some functional importance.

Finally, apical alignment at the RBC membrane is followed by parasite invasion, which requires the formation of a tight junction. During invasion, the tight junction is formed at the apical end and moves toward the back of the parasite with the aid of the actomyosin machinery (*Keeley and Soldati, 2004*; *Robert-Paganin et al., 2019*; *Cowman and Crabb, 2006*). Even though the invasion includes mainy mechanochemical processes, parasite shape must play an important role, as it significantly affects the energy required to deform RBC membrane. For instance, particles with a larger aspect ratio such as oblate and long ellipsoids require a larger energy for complete wrapping (or uptake) (*Bahrami et al., 2014*; *Dasgupta et al., 2014a*; *Dasgupta et al., 2017*). From a dynamical perspective, fluctuations that are important for particle uptake also depend on the geometry of a particle (*Frey et al., 2019*). Note that particle uptake studies are performed majorly for vesicles, while RBCs possess shear elasticity in addition to membrane bending rigidity. For instance, *Hillringhaus et al., 2019* show that for small interaction strengths, bending energy has a dominant contribution to membrane deformation energy, while for strong interactions shear elastic energy exhibits a more dominant contribution. Different aspects related to the performance of various parasite shapes for RBC invasion clearly require further investigations.

## Materials and methods

### Red blood cell model

RBC membrane is modeled as a triangulated surface with $N$ vertices, $N_s$ edges, and $N_t$ faces. The total potential energy is given by *Fedosov et al., 2010a*; *Fedosov et al., 2010b*:

$$U_{\text{rbc}} = U_{\text{sp}} + U_{\text{bend}} + U_{\text{area}} + U_{\text{vol}}. \tag{6}$$

The first term in *Equation 6* represents the elastic energy term $U_{\text{sp}}$ expressed as

$$U_{\text{sp}} = \sum_{i=1}^{N_s} \frac{k_{\text{B}} T \ell_i^{\max} \left(3x_i^2 - 2x_i^3\right)}{4p_{\text{i}}(1 - x_i)} + \frac{\lambda_i}{\ell_i}, \tag{7}$$

where the first term is the worm-like-chain potential, while the second term is a repulsive potential. $\ell_i$ is the length of the i-th spring, $p_i$ is the persistence length, $\ell_i^{\max}$ is the maximum extension, and $x_i = \ell_i/\ell_i^{\max}$. The initial biconcave shape of the RBC is considered to be the stress-free shape, so that it does not have any residual elastic stresses. This is achieved by setting individually all equilibrium spring lengths $\ell_i^0$ to the corresponding edge lengths of the initial membrane triangulation. Shear modulus $\mu$ of the membrane is given in terms of model parameters as *Fedosov et al., 2010a*; *Fedosov et al., 2010b*,

$$\mu = \frac{\sqrt{3}k_{\text{B}}T}{4p_i\ell_i^0} \left(\frac{\bar{x}}{2(1-\bar{x})^3} - \frac{1}{4(1-\bar{x})^2} + \frac{1}{4}\right) + \frac{3\sqrt{3}\lambda_i}{4(\ell_i^0)^3}, \tag{8}$$

where $\bar{x} = \ell_i^0/\ell_i^{\max} = 2.2$ is a constant for all $i$. Thus, for given values of $\mu$, $\bar{x}$, and $\ell_i^0$, individual spring parameters $p_i$ and $\lambda_i$ are calculated by using *Equation (8)* and the force balance $\partial E_{\text{sp}}/\partial l_i|_{l_i^0} = 0$ for each spring.

The second term in *Equation 6* is bending energy of the membrane (*Gompper and Kroll, 1996*; *Gompper and Kroll, 2004*) which is given by

$$U_{\text{bend}} = \frac{\kappa}{2} \sum_{i=1}^{N_{\text{rbc}}} \frac{1}{\sigma_i} \left[\mathbf{n}_i^{\text{rbc}} \cdot \left(\sum_{j(i)} \frac{\sigma_{ij}}{r_{ij}} \mathbf{r}_{ij}\right)\right]^2 \tag{9}$$

where $\kappa$ is the bending modulus, $\mathbf{n}_i^{\text{rbc}}$ is a unit normal of the membrane at vertex $i$, $\sigma_i = \left(\sum_{j(i)} \sigma_{ij} r_{ij}\right)/4$ is the area of dual cell of vertex $i$, and $\sigma_{ij} = r_{ij}[\cot(\theta_1) + \cot(\theta_2)]/2$ is the length of the bond in dual lattice, with the two angles $\theta_1$ and $\theta_2$ opposite to the shared bond $\mathbf{r}_{ij}$.

The last two terms in *Equation (6)* represent surface area and volume constraints,

$$U_{\text{area}} = \frac{k_{\text{a}}(A - A_0)^2}{2A_0} + \sum_{i=1}^{N_t} \frac{k_\ell \left(A_i - A_i^0\right)^2}{2A_i^0}, U_{\text{vol}} = \frac{k_{\text{v}}(V - V_0)^2}{2V_0}. \tag{10}$$

$k_{\text{a}}$ and $k_\ell$ control the total surface area $A$ and local areas $A_i$, while $k_{\text{v}}$ controls the total volume $V$ of the cell. $A_0$ and $V_0$ are total targeted surface area and volume of the cell.

### Parasite model

The parasite is also modeled as a triangulated surface. However, it is treated as a rigid body, as no visual deformations of merozoites are observed in in-vitro experiments (*Weiss et al., 2015*). For all shapes, including the egg-like, sphere, long ellpsoid, short ellipsoid, and oblate shapes, both the surface area and the number of vertices are kept approximately constant, which results in nearly the same density of receptors at the parasite surface. This provides the same adhesion strength between the parasite and RBC membrane for all investigated shapes.

## RBC-parasite interactions

Parasite interacts with the RBC membrane in two ways including excluded volume and adhesion interactions. The excluded-volume interaction is implemented through the Lennard-Jones potential given by

$$U_{\text{rep}}(r) = 4\epsilon\left[\left(\frac{\sigma}{r}\right)^{12} - \left(\frac{\sigma}{r}\right)^{6}\right], \quad r \leq 2^{1/6}\sigma, \tag{11}$$

where $r$ is the distance between RBC and parasite vertices, σ is the repulsive distance chosen to be $0.2\,\mu m$, and $\epsilon = 1000\,k_{\text{B}}T$ is the strength of interaction.

Adhesion interactions are represented by a discrete receptor-ligand bond model. As in our previous work (*Hillringhaus et al., 2020*), two different types of adhesion bonds are used: (*i*) long bonds with an effective length of $\ell_{\text{eff}}^{\text{long}} = 100\,nm$ and (*ii*) short bonds with an effective length of $\ell_{\text{eff}}^{\text{short}} = 20\,nm$. The fraction of long bonds is set to $\rho = 0.4$, while the fraction of short bonds then becomes $1 - \rho = 0.6$. Adhesion bonds between the RBC and the parasite form and dissociate with constant on-rates $k_{\text{on}}^{\text{long}}$ and $k_{\text{on}}^{\text{short}}$, and an off-rate $k_{off}$ which is the same for both bond types. Both long and short bonds are modeled by a harmonic potential as

$$U_{\text{ad}}(\ell) = \frac{\lambda_{\text{type}}}{2}(\ell - \ell_0)^2, \tag{12}$$

where $\lambda_{long}$ and $\lambda_{short}$ are the extensional rigidities of long and short bonds, respectively. $\ell_0 = 2^{1/6}\sigma$ is the equilibrium bond length. Thus, long bonds are formed when the distance between parasite and membrane vertices is less than $\ell_0 + \ell_{\text{eff}}^{\text{long}}$ and short bonds can form when $\ell < \ell_0 + \ell_{\text{eff}}^{\text{short}}$.

## Hydrodynamic interactions

Hydrodynamic interactions are modeled using the dissipative particle dynamics (DPD) method (*Hoogerbrugge and Koelman, 1992*; *Español and Warren, 1995*). DPD models the fluid as a collection of coarse-grained particles which interact through three different pair-wise forces: conservative $\mathbf{F}_{ij}^C$, dissipative $\mathbf{F}_{ij}^D$ and random forces $\mathbf{F}_{ij}^R$. The conservative force which represents the fluid compressibility is given by

$$\mathbf{F}_{ij}^C = a_{ij}\omega^C(r_{ij}), \tag{13}$$

where $a_{ij}$ is the interaction strength, $\omega^C$ is the weight function, and $r_{ij} = r_i - r_j$. The weight function is a decaying function of interparticle distance with a cutoff length $r_c$,

$$\omega^C = \begin{cases} (1 - r_{ij}/r_c), & r\,ij \leq r\,c, \\ 0, & r\,ij > r\,c. \end{cases} \tag{14}$$

The dissipative force $\mathbf{F}_{ij}^D$ and the random force $\mathbf{F}_{ij}^R$ are given by,

$$\mathbf{F}_{ij}^D = -\gamma\omega^D[\mathbf{v}_{ij} \cdot \boldsymbol{e}_{ij}]\boldsymbol{e}_{ij},$$
$$\mathbf{F}_{ij}^R = \sigma\omega^R(r_{ij})\theta_{ij}\boldsymbol{e}_{ij}, \tag{15}$$

where the corresponding weight functions are expressed as,

$$\omega^D = [\omega^R]^2 = \begin{cases} (1 - r_{ij}/r_c)^k, & r\,ij \leq r\,c, \\ 0, & r\,ij > r\,c. \end{cases} \tag{16}$$

Here, both $k$ and the dissipative coefficient γ control the viscosity of DPD fluid. $\theta_{ij}$ is a white noise with zero mean and unit variance. The fluctuation-dissipation theorem connects both σ and γ as $\sigma^2 = 2\gamma k_{\text{B}}T/m$(*Español and Warren, 1995*). The DPD interactions are implemented between fluid-fluid, fluid-RBC vertices and fluid-parasite vertices, but not between RBC-parasite vertices. The dissipative coefficient is always chosen to make sure no-slip boundary conditions are satisfied (*Fedosov et al., 2010a*; *Hillringhaus et al., 2019*).

## Simulation setup

All simulations are carried out in a simulation domain of size $7.7D_0 \times 3.1D_0 \times 3.1D_0$ with periodic boundary conditions in all directions, where $D_0 = \sqrt{A_0/\pi}$ is the effective RBC diameter and $A_0$ is the membrane area. Receptors for both long and short bonds are chosen randomly over the parasite surface and this procedure is repeated for every realization to obtain a good averaging of physical quantities. In every simulation, the parasite is placed close to the RBC membrane in order to facilitate its initial attachment. The initial position of the parasite is with its back toward the membrane, so that its apex is directed away from the membrane. In all simulations, the initial position is fixed. RBC bending rigidity is chosen to be $\kappa = 3 \times 10^{-19}$ J and $A_0 = 133\,\mu m^2$ resulting in $D_0 = 6.5\,\mu m$. Fluid viscosity inside and outside the RBC is set to $\eta = 1\,mPa.s$. To connect simulation and physical units, we use $D_0$ as a length scale, $k_BT$ as an energy scale, and the RBC membrane relaxation time $\eta D_0^3/\kappa$ as a time scale. All simulations were performed on JURECA, a super-computer at Forschungszentrum Jülich *Krause and Thörnig, 2018*.

## Calibration of bond kinetic parameters

Kinetic parameters for the adhesion between the RBC and parasite are tuned such that the parasite displacement from simulations matches well the merozoite displacement from the experimental data (*Weiss et al., 2015*). The calibration is performed for the egg-like shape and the resultant 'reference' parameters given in *Table 1* are used for all other parasite shapes. The detailed procedure is explained in *Hillringhaus et al., 2020*.

## Alignment times: Monte Carlo sampling

A Monte Carlo sampling scheme is employed for measuring alignment times from probability maps of parasite alignment characteristics (apex distance $d_{apex}$ and alignment angle θ), which are constructed from approximately 10 independent long simulations for each parameter set. Briefly, the Monte Carlo procedure is as follows. First, a state $(i, j)$ is randomly selected, which corresponds to specific $(d_{apex}^i, \theta_j)$ values in a probability map. Second, a transition to one of the four neighboring states with a probability of 0.25 is attempted, and it is accepted if $\zeta < P(\mathrm{newstate})/P(i, j)$ where $\zeta$ is a uniform random number. This state transition is repeated until a state that meets the alignment criteria is reached. Then, the total alignment time is equal to the total number of Monte Carlo moves, see *Hillringhaus et al., 2020* for more details. All alignment times are normalized by the corresponding time for the reference parameter set, which is obtained through the calibration of parasite speed (*Hillringhaus et al., 2020*) against available experimental data (*Weiss et al., 2015*).

## Acknowledgements

Sebastian Hillringhaus acknowledges support by the International Helmholtz Research School of Biophysics and Soft Matter (IHRS BioSoft). We gratefully acknowledge the computing time granted through JARA-HPC on the supercomputer JURECA Jülich Supercomputing Centre (2018) at Forschungszentrum Jülich.

## Additional information

### Funding

| Funder | Author |
| --- | --- |
| International Helmholtz Research School of Biophysics and Soft Matter | Sebastian Hillringhaus |

The funder had no role in study design, data collection and interpretation, or the decision to submit the work for publication.

## Author contributions
Anil K Dasanna, Software, Formal analysis, Investigation, Visualization, Methodology, Writing - original draft; Sebastian Hillringhaus, Software, Formal analysis, Investigation, Methodology; Gerhard Gompper, Conceptualization, Project administration, Writing - review and editing; Dmitry A Fedosov, Conceptualization, Supervision, Project administration, Writing - review and editing

## Author ORCIDs
Anil K Dasanna (ID) https://orcid.org/0000-0001-5960-4579
Sebastian Hillringhaus (ID) http://orcid.org/0000-0003-0100-9368
Gerhard Gompper (ID) https://orcid.org/0000-0002-8904-0986
Dmitry A Fedosov (ID) https://orcid.org/0000-0001-7469-9844

## Decision letter and Author response
Decision letter https://doi.org/10.7554/eLife.68818.sa1
Author response https://doi.org/10.7554/eLife.68818.sa2

# Additional files

## Supplementary files
• Transparent reporting form

## Data availability
All data generated or analysed during this study are included in the manuscript and supporting files. Source data for all figures are provided.

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
