## [Decision Letter]

**Acceptance summary:**

This manuscript studies the alignment of malaria parasites (merozoites) at the surface of red blood cells (RBCs), a key element of their reproduction cycle during the blood stage of the disease. Building on a computational model the authors developed previously that incorporates the stochastic nature of RBC deformations and adhesive bonds between the merozoite and RBC, it is demonstrated that parasite shape plays a key role in its alignment dynamics. The authors shed new light on the egg-like shape typically observed in Plasmodium merozoites, which has important implications for how effectively the parasite can survive and multiply.

**Decision letter after peer review:**

Thank you for submitting your article "Effect of malaria parasite shape on its alignment at erythrocyte membrane" for consideration by *eLife*. Your article has been reviewed by 2 peer reviewers, and the evaluation has been overseen by a Reviewing Editor and Suzanne Pfeffer as the Senior Editor. The following individual involved in review of your submission has agreed to reveal their identity: Michael Gomez (Reviewer #1).

Essential revisions:

1. Figure 1a: It would be helpful to plot the egg-like shape here too, so it can be easily compared to the other shapes (including the short-ellipsoid, to which it is most similar). Also, highlighting the apex on each shape would be helpful. (There is also the typo "elliposid".)

2. Line 106: It would be good to refer to the details of the computational model in the Methods section at this point.

3. Line 110 (and Figures 1c, 2a-b, 3a, 4a, 5a): Is there a particular reason why the effective RBC diameter D_0_ is used to normalize the fixed-time displacement Δd and apex distance d_apex_ when presenting the results? Since the RBC is much larger than the parasite, this means the normalized values are all much smaller than unity. A more informative choice might be to normalize by the effective merozoite diameter, equal to the square root of A_s_/π where A_s_ is the typical merozoite surface area (which is precisely 2R in the case of a sphere). The normalized values of Δd would then give a better indication of how much the merozoites move relative to their size, and the normalized values of d_apex_ would lie in the range [0,1].

4. Lines 131-140: It would be helpful to have a schematic illustrating the quantities n, n_face_, d_apex_ and θ, similar to Figure 3a in Hillringhaus et al., 2020. There should also be a brief description of where the alignment criteria in Equation 2 come from (in addition to referencing Hillringhaus et al., 2020), including the meaning of the 2^1/6^σ term and that d_apex_ cannot obtain values below the repulsion length.

5. Lines 145-146: The authors should be more precise here as to what features of the alignment-angle distributions make the egg-like shape align better than the LE and OB shapes. The LE and OB shapes have a narrower distribution with a peak at θ/π ≈ 0.6, which presumably corresponds to the configuration of largest adhesion area in which the apex is pointing almost tangentially to the membrane. The tapering of the egg-like shape breaks the fore-aft symmetry and tilts the apex towards the membrane in the configuration of largest adhesion area.

6. Lines 146-150 and Figures 2a,c: It is worth emphasising here that membrane deformability is what breaks the rotational symmetry for a spherical shape, so that the alignment-angle distribution is not uniform: if the apex is within the contact area A_m_, then the deformation of the membrane will push θ closer to π.

7. Line 156: The reference to the inset of Figure 3a here is confusing, since the situation sketched there (with the apex away from the contact region) is not what is being discussed in the text. It might be helpful to have a sketch of both scenarios (i.e. apex inside and outside the contact area), which could be combined with the new schematic suggested in comment 4 above.

8. Lines 177-189: The area being described in Equation 3 is not quite clear: it is worth mentioning the area is what is called a spherical segment/frustum, whose curved surface area depends only on the sphere radius and the height h of the segment (not its slanted length).

9. Figure 4a: The label on the y-axis should be d_apex_/D_0_.

10. Lines 260-261: Is it possible to speculate as to why the merozoite of *Plasmodium yoelii* changes its shape from oval to spherical prior to attachment to a RBC membrane? For example, could it be that it has much weaker adhesive bonds (consistent with the observation that the alignment times are longer than those of *Plasmodium falciparum*), so that it is always in the high-mobility/low-adhesion regime in which a spherical shape is advantageous?

11. A primary concern with this manuscript is the lack of detail on what is done in the simulations. While we know that the authors have published a detailed description of the method elsewhere, it is not sufficient to just cite that, requiring a reader to read a separate paper to even get a basic understanding of what goes on in the simulations. While we understand that the authors do not want to provide the full description here, they need to provide enough description that a reader is aware of the basic implementation and can look to the other publication to fill in details, if needed.

12. As is, the authors state that the RBC and the merozoite are handled using triangulated meshes, that adhesion molecules are located at the vertices of these meshes, and that there is an external fluid that is simulated using dissipative particle dynamics. How is the elasticity of the RBC handled? How is the rigidity of the merozoite handled? We would imagine that the alignment time, etc depends on where on the RBC the merozoite is, since the RBC membrane does not have constant curvature. Are all simulations started from the same location on the RBC surface or are these randomly assigned? How might either of these choices affect the results? Basic parameters of the DPD should also be included.

*Reviewer #1:*

This manuscript studies the invasion of red blood cells (RBCs) by malaria parasites (merozoites), a key element of their reproduction cycle during the blood stage of the disease. For successful invasion to occur, the merozoite must first align its apex almost perpendicularly to the RBC membrane. In a previous study (reference Hillringhaus et al., 2020), the authors developed a computational model that incorporates stochastic deformations of the RBC membrane and the discrete nature of the adhesive bonds between the merozoite and RBC (arising from filaments on the merozoite surface); together these effects enable partial wrapping of the membrane around the merozoite to aid alignment. This manuscript builds upon this framework to examine the influence of the parasite shape on the alignment dynamics, using five different reference shapes: the egg-like shape typical of Plasmodium merozoites, a sphere, and ellipsoids of varying aspect ratio. By exploring the influence of various parameters such as the bond kinetics and RBC membrane stiffness, they demonstrate that the parasite shape plays a key role in its alignment dynamics. In particular, the egg-like shape is found to be more robust to different adhesion strengths and membrane deformability: it is relatively mobile compared to the ellipsoidal shapes and, unlike a sphere, does not easily become arrested in the high-adhesion limit due to its lack of spherical symmetry.

The manuscript is excellently written and discusses the simulation results clearly and succinctly. The resolution of the simulations is very impressive and yields unprecedented insight into the effect of merozoite shape on alignment dynamics, which has important implications for how effectively the parasite can survive and multiply. The conclusions reached by the authors are certainly justified by the simulation data. In particular, the authors are careful not to draw conclusions beyond the limits of their study, and acknowledge other factors which may influence the merozoite shape, such as internal structural constraints and the energy of invasion following successful alignment.

Regarding weaknesses of the manuscript, some of the explanations of the trends observed in the simulation data could be expanded slightly, to help gain a deeper understanding of the competition between adhesion and RBC deformability underlying the alignment dynamics. These are described in more detail below.

1. Line 114 and lines 120-129: The discussion here of the trends observed in Figure 1 (including why the LE shape has a larger energy compared to the OB shape despite having a smaller adhesion area) is somewhat vague and should be developed further. For example, currently there is only a video showing the egg-like shape and a second video comparing the LE shape to a spherical shape – it would be helpful to have a further video comparing the LE and OB shapes and the different RBC deformations they cause. Moreover, the explanation of the energy/mobility of each shape in terms of curvatures (e.g. the OB shape having "lower curvature at its flat side") could be made more precise. I would expect that the adhesion area depends on how close the principal curvatures of the merozoite surface are to being equal and opposite to the natural curvatures of the RBC, since this determines the bending energy associated with wrapping the merozoite and forming short bonds. This would explain why the spherical shape is most mobile (its principal curvatures are constant so there is no region where at least one is relatively small), and why alignment is most likely to occur in the dimple of the RBC where the membrane is naturally concave-outward. For a given adhesion area, the deformation energy should depend on the difference in principal curvatures in the contact region, with a larger difference causing more bending of the RBC membrane. This difference is larger for the LE shape, since one principal curvature remains large at each point on the surface, compared to the OB shape whose principal curvatures are both small on the 'flat side' where contact is most likely to occur.

2. Lines 175-176: Given that the ratio A_m_/A_s_ (adhesion area to total surface area) plays a key role in the probability of alignment, the authors should be more quantitative at this point. How does the ratio A_m_/A_s_ (as measured directly, or indirectly e.g. by the area under the probability distributions inside the alignment region in figures 3a,b) scale with the system parameters, such as the adhesion strength and the off-rate k_off_? Can it be estimated from an energy balance between RBC bending/stretching and the average adhesion energy?

3. Line 197-198 and Figure 4c: Why is the deformation energy associated with the OB shape much lower than all other shapes for values of koff/konlong<2?4. Alignment requires that the distance between the merozoite apex and RBC membrane is very small, and the alignment criteria necessitate examining small changes in the apex angle \theta from \pi. Can the authors comment on how sensitive are the results to the numerical discretisation used?

*Reviewer #2:*

This manuscript seeks to determine the role that malarial shape plays in the ability of this parasite to infect red blood cells. The authors use computational modeling to explore the dynamics of different parasite shapes and the effect of adhesion strength in getting the malaria parasite to bind into the correct orientation for invasion into the red blood cell.

A major strength of the results is that it investigates an unstudied problem in malarial pathogenesis. The results pertaining to adhesion strength may be informative for preventing the organism from invading red blood cells. A primary weakness is that there is too little detail provided in the methods for this reviewer to adequate assess the computational method. Secondly, the results are somewhat inconclusive. While the egg-shape performs better than certain other shapes, there is no clear final understanding why this shape is preferred over the spherical or short ellipsoidal shapes. However, this possibly provides some clues as to why a certain malarial species does actively adopt a spherical shape during red blood cell binding and invasion.

Overall, the authors achieved their aims by quantitatively assessing the effect of parasite shape and adhesion strength on cell alignment, which is a proxy for invasion. The discussion at the end of the manuscript provides an accurate evaluation of the results that puts them into the context of invasion.

While to some extent the results presented here are inconclusive, I do think that this paper achieves an important goal for its field. This is an understudied area pertinent to a major disease. This manuscript has the potential to bring questions of the biophysics of malarial invasion out to the broader community, specifically introducing these questions to biophysicists as well as microbiologists. Furthermore, the results naturally lead to new questions. If the spherical and egg shapes do not confer a strong advantage, then these specific shapes must also play a role in other processes. The authors do suggest some possibilities in the Discussion. That their remain interesting questions is a great spur for future work.

---

## [Author Response]

Essential revisions:1. Figure 1a: It would be helpful to plot the egg-like shape here too, so it can be easily compared to the other shapes (including the short-ellipsoid, to which it is most similar). Also, highlighting the apex on each shape would be helpful. (There is also the typo "elliposid".)

We have modified Figure 1 and included the egg-like shape along with the other shapes. The position of the apex is also indicated now for all shapes. We thank the reviewer for spotting the typo, which has been corrected.

2. Line 106: It would be good to refer to the details of the computational model in the Methods section at this point.

We have added a reference to the Methods section there.

3. Line 110 (and Figures 1c, 2a-b, 3a, 4a, 5a): Is there a particular reason why the effective RBC diameter D_0_ is used to normalize the fixed-time displacement Δd and apex distance d_apex_ when presenting the results? Since the RBC is much larger than the parasite, this means the normalized values are all much smaller than unity. A more informative choice might be to normalize by the effective merozoite diameter, equal to the square root of A_s_π where A_s_ is the typical merozoite surface area (which is precisely 2R in the case of a sphere). The normalized values of Δd would then give a better indication of how much the merozoites move relative to their size, and the normalized values of d_apex_ would lie in the range [0,1].

We agree with the reviewer that a normalization with the parasite size might be more intuitive for these quantities. Therefore, both the translational displacement and apex distance are normalized now with the merozoite’s diameter instead of RBC diameter and the corresponding figures and text were modified. We have used the surface area of an egg-like shape for the normalization of both quantities for all shapes, as they have approximately the same surface area. As expected, now the major part of d_apex_ distributions is mainly within the range of [0,1]. Note that d_apex_ normalized with the merozoite diameter can take values slightly larger than unity.

4. Lines 131-140: It would be helpful to have a schematic illustrating the quantities n, n_face_, d_apex_ and θ, similar to Figure 3a in Hillringhaus et al., 2020. There should also be a brief description of where the alignment criteria in Equation 2 come from (in addition to referencing Hillringhaus et al., 2020), including the meaning of the 2^1/6^σ term and that d_apex_ cannot obtain values below the repulsion length.

We have added both a sketch and the corresponding text, providing more details on the alignment characteristics and criterion in the manuscript.

5. Lines 145-146: The authors should be more precise here as to what features of the alignment-angle distributions make the egg-like shape align better than the LE and OB shapes. The LE and OB shapes have a narrower distribution with a peak at θ/π ≈ 0.6, which presumably corresponds to the configuration of largest adhesion area in which the apex is pointing almost tangentially to the membrane. The tapering of the egg-like shape breaks the fore-aft symmetry and tilts the apex towards the membrane in the configuration of largest adhesion area.

We completely agree with the reviewer about this argument. We have improved the discussion of figures 2 (b) and (d) to make this point clear.

6. Lines 146-150 and Figures 2a,c: It is worth emphasising here that membrane deformability is what breaks the rotational symmetry for a spherical shape, so that the alignment-angle distribution is not uniform: if the apex is within the contact area A_m_, then the deformation of the membrane will push θ closer to π.

Thank you for emphasizing this. The relevant text has been added to the manuscript.

7. Line 156: The reference to the inset of Figure 3a here is confusing, since the situation sketched there (with the apex away from the contact region) is not what is being discussed in the text. It might be helpful to have a sketch of both scenarios (i.e. apex inside and outside the contact area), which could be combined with the new schematic suggested in comment 4 above.

We have added two sketches of a spherical parasite, interacting with the membrane and representing the two cases of apex orientation, namely within the adhesion area and away from it. These sketches are combined with the sketch that the referee suggested in question 4.

8. Lines 177-189: The area being described in Equation 3 is not quite clear: it is worth mentioning the area is what is called a spherical segment/frustum, whose curved surface area depends only on the sphere radius and the height h of the segment (not its slanted length).

We mention it now explicitly in the text.

9. Figure 4a: The label on the y-axis should be d_apex_/D_0_.

We have corrected it.

10. Lines 260-261: Is it possible to speculate as to why the merozoite of *Plasmodium yoelii* changes its shape from oval to spherical prior to attachment to a RBC membrane? For example, could it be that it has much weaker adhesive bonds (consistent with the observation that the alignment times are longer than those of *Plasmodium falciparum*), so that it is always in the high-mobility/low-adhesion regime in which a spherical shape is advantageous?

We are not aware any data which would directly support a lower adhesion of Plasmodium yoelii in comparison to *Plasmodium falciparum*. Nevertheless, we think that this is an interesting idea and we have added discussion about a possible advantage of the spherical shape for Plasmodium yoelii.

11. A primary concern with this manuscript is the lack of detail on what is done in the simulations. While we know that the authors have published a detailed description of the method elsewhere, it is not sufficient to just cite that, requiring a reader to read a separate paper to even get a basic understanding of what goes on in the simulations. While we understand that the authors do not want to provide the full description here, they need to provide enough description that a reader is aware of the basic implementation and can look to the other publication to fill in details, if needed.

We have significantly expanded the Materials and methods section, which should now contain all necessary model details.

12. As is, the authors state that the RBC and the merozoite are handled using triangulated meshes, that adhesion molecules are located at the vertices of these meshes, and that there is an external fluid that is simulated using dissipative particle dynamics. How is the elasticity of the RBC handled? How is the rigidity of the merozoite handled? We would imagine that the alignment time, etc depends on where on the RBC the merozoite is, since the RBC membrane does not have constant curvature. Are all simulations started from the same location on the RBC surface or are these randomly assigned? How might either of these choices affect the results? Basic parameters of the DPD should also be included.

The modeled RBC has an in-plane shear elasticity and bending resistance, while the merozoite is treated as a rigid body. We have added necessary details to the Materials and methods section. Details about the DPD method have also been added to the Materials and methods section.

For all simulations, we initially place the parasite close to RBC surface (at the side) with the back side of the parasite (i.e. the apex is oriented away from the membrane). The alignment time indeed depends on the initial placement, which has been discussed in our previous article Hillringhaus et al. *eLife* (2020). In this manuscript, our main focus was to look at the effect of the parasite shape, which already leaves us with a relatively large parameter space.

Reviewer #1:al.[…] Regarding weaknesses of the manuscript, some of the explanations of the trends observed in the simulation data could be expanded slightly, to help gain a deeper understanding of the competition between adhesion and RBC deformability underlying the alignment dynamics. These are described in more detail below.1. Line 114 and lines 120-129: The discussion here of the trends observed in Figure 1 (including why the LE shape has a larger energy compared to the OB shape despite having a smaller adhesion area) is somewhat vague and should be developed further. For example, currently there is only a video showing the egg-like shape and a second video comparing the LE shape to a spherical shape – it would be helpful to have a further video comparing the LE and OB shapes and the different RBC deformations they cause. Moreover, the explanation of the energy/mobility of each shape in terms of curvatures (e.g. the OB shape having "lower curvature at its flat side") could be made more precise. I would expect that the adhesion area depends on how close the principal curvatures of the merozoite surface are to being equal and opposite to the natural curvatures of the RBC, since this determines the bending energy associated with wrapping the merozoite and forming short bonds. This would explain why the spherical shape is most mobile (its principal curvatures are constant so there is no region where at least one is relatively small), and why alignment is most likely to occur in the dimple of the RBC where the membrane is naturally concave-outward. For a given adhesion area, the deformation energy should depend on the difference in principal curvatures in the contact region, with a larger difference causing more bending of the RBC membrane. This difference is larger for the LE shape, since one principal curvature remains large at each point on the surface, compared to the OB shape whose principal curvatures are both small on the 'flat side' where contact is most likely to occur.

We have expanded the discussion of these results to make it clearer. Furthermore, a new video was generated to visually see differences between different shapes.

2. Lines 175-176: Given that the ratio A_m_/A_s_ (adhesion area to total surface area) plays a key role in the probability of alignment, the authors should be more quantitative at this point. How does the ratio A_m_/A_s_ (as measured directly, or indirectly e.g. by the area under the probability distributions inside the alignment region in figures 3a,b) scale with the system parameters, such as the adhesion strength and the off-rate k_off_? Can it be estimated from an energy balance between RBC bending/stretching and the average adhesion energy?

A change in A_m_ as a function of adhesion strength can be estimated analytically for a sphere, as was done in Hillringhaus et al. Biophys. J. 117:1202, 2019. For small deformations, there is essentially a competition of bending and adhesion energies, while for strong adhesion, stretching-elasticity contribution becomes important. We have included this theoretical result into the manuscript and discuss its implications.

3. Line 197-198 and Figure 4c: Why is the deformation energy associated with the OB shape much lower than all other shapes for values of koff/konlong<2?

For koff/konlong<2, the magnitude of local curvature has a pronounced effect. For the OB shape, a large adhesion area is formed over the area with very low curvature, and close to the rim where the curvature is large, the adhesion strength may not be strong enough to induce membrane wrapping and deformation. For other shapes, the adhesion strength is large enough to lead to partial wrapping of the parasite by the membrane over moderate curvatures. As a result, the integrated deformation energy is significantly lower for the OB shape than for the other shapes in this regime of adhesion strengths. We have added this clarification to the manuscript.

4. Alignment requires that the distance between the merozoite apex and RBC membrane is very small, and the alignment criteria necessitate examining small changes in the apex angle \theta from \pi. Can the authors comment on how sensitive are the results to the numerical discretisation used?

The discretization length does affect the tightness of the alignment criteria. In our simulations, the average discretization length of the RBC membrane is about l_0_=0.2 μm. The half circumference length of a parasite (corresponding to angle π) is πR, which is equal to about 12 l_0_ for R=0.75 μm, such that our angle resolution with respect to the parasite size is 0.1π. Therefore, we use 0.2π for the alignment criteria, which is large enough to avoid strong discretization effects. Simulations with a finer discretization are possible, but they become very expensive computationally.

Reviewer #2:This manuscript seeks to determine the role that malarial shape plays in the ability of this parasite to infect red blood cells. The authors use computational modeling to explore the dynamics of different parasite shapes and the effect of adhesion strength in getting the malaria parasite to bind into the correct orientation for invasion into the red blood cell.A major strength of the results is that it investigates an unstudied problem in malarial pathogenesis. The results pertaining to adhesion strength may be informative for preventing the organism from invading red blood cells. A primary weakness is that there is too little detail provided in the methods for this reviewer to adequate assess the computational method. Secondly, the results are somewhat inconclusive. While the egg-shape performs better than certain other shapes, there is no clear final understanding why this shape is preferred over the spherical or short ellipsoidal shapes. However, this possibly provides some clues as to why a certain malarial species does actively adopt a spherical shape during red blood cell binding and invasion.

We thank the reviewer for a positive judgment of our manuscript. We have significantly expanded the methods section, so it should contain now all necessary simulation details. We agree with the reviewer that the conclusions about shape advantages/disadvantages are equivocal to some extent, but this is exactly what our simulation data show. However, from our data it is clear that the two shapes (i.e. egg-like and sphere) stand out, and they also correspond to real examples of merozoite shapes. As the reviewer points out, we do discuss some clues for the importance of parasite shape in the alignment process.

Overall, the authors achieved their aims by quantitatively assessing the effect of parasite shape and adhesion strength on cell alignment, which is a proxy for invasion. The discussion at the end of the manuscript provides an accurate evaluation of the results that puts them into the context of invasion.While to some extent the results presented here are inconclusive, I do think that this paper achieves an important goal for its field. This is an understudied area pertinent to a major disease. This manuscript has the potential to bring questions of the biophysics of malarial invasion out to the broader community, specifically introducing these questions to biophysicists as well as microbiologists. Furthermore, the results naturally lead to new questions. If the spherical and egg shapes do not confer a strong advantage, then these specific shapes must also play a role in other processes. The authors do suggest some possibilities in the Discussion. That their remain interesting questions is a great spur for future work.

Thank you for emphasizing the importance of multidisciplinarity. We also hope that our work will ignite interest in different communities, as only a multidisciplinary effort can bring us much closer to understanding of parasite alignment and invasion, which clearly include a combination of different mechanical and biochemical processes.